# Antimicrobial Resistance in Loggerhead Sea Turtles (*Caretta caretta*): A Comparison between Clinical and Commensal Bacterial Isolates

**DOI:** 10.3390/ani11082435

**Published:** 2021-08-18

**Authors:** Adriana Trotta, Mariarosaria Marinaro, Alessio Sposato, Michela Galgano, Stefano Ciccarelli, Serena Paci, Marialaura Corrente

**Affiliations:** 1Department of Veterinary Medicine, University of Bari “Aldo Moro”, Strada Provinciale per Casamassima Km 3, 70010 Valenzano, Italy; alessio.sposato@uniba.it (A.S.); michela.galgano@uniba.it (M.G.); stefano.ciccarelli@uniba.it (S.C.); serena.paci@uniba.it (S.P.); marialaura.corrente@uniba.it (M.C.); 2Department of Infectious Diseases, Istituto Superiore di Sanità, Viale Regina Elena 299, 00161 Rome, Italy; mariarosaria.marinaro@iss.it

**Keywords:** sea turtles, infectious diseases, sentinel species, marine ecosystem, AMR, pathogens

## Abstract

**Simple Summary:**

Gram negative organisms are frequently isolated from *Caretta caretta* and may contribute to the dissemination of antimicrobial resistance. In this study, commensal bacteria isolated from oral and cloacal samples of 98 healthy *C. caretta* were compared to clinical isolates isolated from the wounds of 102 injured animals, in order to investigate the presence of antimicrobial resistance bacteria in free-living loggerheads from the Adriatic Sea. A total of 410 bacteria were cultured and differences were noted in the isolated genera, as some of them were isolated only in healthy animals, while others were isolated only from injured animals. When tested for susceptibility to antimicrobials, clinical isolates showed highly significant differences in the antimicrobial resistance rates vs. commensal isolates for all the drugs tested, except for doxycycline. The detection of high antimicrobial resistance rates in loggerhead sea turtles is of clinical and microbiological significance since it impacts both the choice of a proper antibiotic therapy and the implementation of conservation programs.

**Abstract:**

Gram negative organisms are frequently isolated from *Caretta caretta* turtles, which can act as reservoir species for resistant microorganisms in the aquatic environment. *C. caretta,* which have no history of treatment with antimicrobials, are useful sentinel species for resistant microbes. In this culture-based study, commensal bacteria isolated from oral and cloacal samples of 98 healthy *C. caretta* were compared to clinical isolates from the wounds of 102 injured animals, in order to investigate the presence of AMR bacteria in free-living loggerheads from the Adriatic Sea. A total of 410 isolates were cultured. *Escherichia coli* and genera such as *Serratia, Moraxella, Kluyvera, Salmonella* were isolated only in healthy animals, while *Acinetobacter, Enterobacter, Klebsiella and Morganella* were isolated only from the wounds of the injured animals. When tested for susceptibility to ampicillin, amoxicillin + clavulanic acid, ceftazidime, cefuroxime, gentamicin, doxycycline, ciprofloxacin and enrofloxacin, the clinical isolates showed highly significant differences in AMR rates vs. commensal isolates for all the drugs tested, except for doxycycline. The detection of high AMR rates in loggerheads is of clinical and microbiological significance since it impacts both the choice of a proper antibiotic therapy and the implementation of conservation programs.

## 1. Introduction

The overuse and misuse of antibiotics in human and veterinary medicine, as well as in agriculture and aquaculture, has contributed to increasing the selective pressure on bacteria, with the consequent appearance of new resistant strains and the dissemination of antimicrobial resistance (AMR) in various environments [1,2].

The extensive use of antimicrobials has also caused their dispersal to sanitary sewers, landfills and near the coast [1]. It is estimated that up to 90% of all wastewater is discharged, untreated, directly into rivers, lakes or oceans [3] and, for this reason, AMR determinants could be used as indicators of pollution in marine ecosystems [4]. In addition, resistant microorganisms can persist in the aquatic environment in reservoir species [4,5,6] and some wild marine species, with no history of treatment with antimicrobials, can harbour resistant microbial species responsible for AMR genes recombination [4,7,8,9,10]. According to a recent 20-year study conducted by the French Research Institute for Exploitation of the Sea (Ifremer), the Mediterranean marine water is the most waste-polluted in Europe [11]. In particular, the accumulation of AMR organisms in marine sediments and animals involves the coastal seawater with shallow bottoms, such as the Adriatic coast [11,12]. Moreover, the pollution of the Adriatic Sea has been also indirectly investigated by using wild “sentinel species” such as fish and mussels [7,13] and by studying fish from aquaculture [5,14,15].

The European Union (EU) Marine Strategy Framework Directive (Commission’s Decision 2010/447/EU) supports research on loggerhead sea turtles (*Caretta caretta*) microbiota in order to study the impact of marine litter [15]. Loggerhead sea turtles have no history of antibiotic therapy and, for their ecological and physiological characteristics, tend to bioaccumulate contaminants such as toxins, pathogens and heavy metals [16,17,18]. In addition, high site fidelity in the adult and juvenile life stages makes this species a perfect candidate for sentinel programs in the Adriatic Sea [19]. Sea turtles are highly impacted by human activities in marine environments; in fact, both direct (trawling and nests destruction) [10,18,20,21] and indirect (macroplastic waste and the dispersion of pesticides/heavy metals) [17,22] activities have resulted in the inclusion of this species in the IUCN Red List (http://www.iucnredlist.org/details/3897/0, accessed on 23 April 2020). 

Bacterial infections caused by Gram negative opportunistic pathogens are commonly reported in free-ranging and captive sea turtles [23,24,25,26,27]. Many of these microorganisms are part of the sea turtle microbiota but can be dangerous, not only due to their opportunistic behaviour but also as a result of the possible dissemination of resistance genes, such as those conferring resistance to β-lactams, to other marine bacteria [26,28,29,30]. Indeed, culture-based studies on the antimicrobial susceptibility of microbial communities isolated from marine turtles [26,31,32,33] have suggested that Gram negative isolates are often inherently resistant to several drugs, but are susceptible to last generation penicillin and cephalosporins, aminoglycosides and fluoroquinolones [27,33].

A few studies have compared the bacteria sampled from free-living and hospitalized loggerheads or have documented the presence of AMR isolates from cloacal and oral swabs of free-living turtles [27,31,33]. Additionally, in the Mediterranean area, most of the studies have focused on AMR bacteria in loggerheads living in the western side of the Mediterranean Sea [4,26,34] creating a gap in the surveillance of loggerheads living in the Adriatic Sea.

To fill this gap, the current study was designed to identify and compare the microbial communities isolated from oral and cloacal (O/C) samples of healthy *C. caretta* with those obtained from the wounds of injured animals. In particular, a large number of bacteria isolated from *C. caretta,* living only in the Adriatic Sea, was collected, identified and compared over a period of 4 years.

## 2. Materials and Methods

### 2.1. Turtles and Specimen Collection

The animals included in the report were all housed in the “Sea Turtle Clinic” (STC) of the DVM University of Bari (Italy) and the entire study lasted 4 years (2018–2021). All the turtles were found stranded on the beach or they were accidentally captured by local fishermen, therefore, they were brought to the STC by the volunteers of the “Centro Recupero tartarughe marine WWF Molfetta” for routine checks. In order to assess the health status of the subjects (healthy subject/subject with wounds) each animal was routinely examined by veterinary personnel, upon their arrival at the STC. In addition, a radiographical investigation was performed to check the pulmonary status. All the animals were hospitalized in the STC for a variable period of time (from 24 h for healthy subjects or until complete recovery for unhealthy subjects, which required weeks to months) and then released into the sea. For the study, all turtles were sampled as soon as possible. During the survey, only untagged turtles were sampled, as presumptively none of them had previously received antibiotics.

For the age classification, curved carapace length (CCL) and weight data were collected from each animal. In particular, subjects with less than 60 cm CCL and less than 25 kg weight were classified as juvenile, subjects with more than 60 cm CCL and more than 25/30 kg weight but showing no sexual dimorphisms were classified as subadult, while subjects ready for reproduction and showing sexual characters were classified as adult [35,36].

A total of 200 juvenile, subadult and adult loggerhead sea turtles (*C. caretta*) found stranded or captured by local fishermen, from the western Adriatic Sea of Apulia, Italy (Figure 1), were included in the study.

Signalling data of turtles are reported in Table 1 and, due to the large number of subjects, they were divided into 3 subgroups based on their life stage (i.e., juvenile, subadult and adult).

Animals were restrained by hand during collection of specimens and no anaesthesia was used. Sterile swabs immersed in transport media (Nuova APTACA, Brescia, Italy) were used for the bacteriological analyses.

From each of the 98 healthy turtles, 1 oral and 1 cloacal swab (O/C) were collected in order to culture the commensal microorganisms. For oral samples, the turtle beak and mouth were opened by hand without the use of additional tools and swab was gently rotated on tongue and palate mucosa. For cloacal samples, swab was gently inserted and rolled inside the cloaca (10 cm internal depth). 

From the 102 injured turtles, swabs were collected from external or internal wounds, in order to investigate the associated microorganisms and to select the proper antibiotic therapy. External samples, such as those from skin carapace or plastron wounds, were collected during clinical examination, after superficial curettage; while internal samples, i.e., those from organ injuries, such as internal biopsies or bronchioalveolar lavage (BAL) (20 mL), were collected during surgery. In particular, extensive, deep and acute lesions from external tissues were sampled, while chronic and/or partially healed wounds were not sampled. Internal biopsies were selected for bacteriological investigation only when clinicians suspected a bacterial infection. BAL was collected from turtles with suspected or diagnosed (by radiographical exams) pulmonary diseases. Details of the type of injury in turtles are reported in Appendix A.

After collection, samples were immediately transported to the bacteriology laboratory of the DVM.

### 2.2. Microbiological Analysis

Swabs were cultured on 3 different agar media and 1 broth: 5% Columbia blood agar (CBA), McConkey agar (MCK), Mannitol salt agar (MSA), and Tryptic Soy broth (TSB), (Liofilchem, Teramo, Italy). The media were incubated in aerobic condition at 35 °C for 24–48 h. Initial presumptive bacterial species identification was performed by Gram staining technique. Bacteria grown only on CBA were subjected to Gram staining identification. Bacteria grown on MCK, i.e., lactose positive or lactose negative on MCK, were tested with the oxidase test (Liofilchem, Teramo, Italy) and subjected to biochemical identification using micro-method tests API 20E and API 20NE (Biomérieux, France).

For the isolation of Salmonellae from oral and cloacal samples, a three-step procedure was used [37]. Briefly, each sample was inoculated into Buffered Peptone Water (BPW, Oxoid, Italy) and incubated at 37 °C, 1 mL of BPW was transferred after 48 h of incubation into 9 mL of Rappaport Vassiliadis broth (RVB, Liofilchem, Italy) and incubated at 42 °C. After 24 h of incubation of RVB the samples were cultured on Xylose Lysine Desoxycholate Agar (XLD, Liofilchem, Italy). All the plates were incubated at 37 °C for 24 h.

Identification of the colonies grown on XLD agar with properties typical of *Salmonella* spp., as indicated by the supplier’s instructions, were selected from each plate and tested using a genus-specific PCR targeting the invA gene [38].

The 16S rRNA gene amplification and sequencing were used for the isolates not successfully identified with biochemical tests. Details of the PCR are described in Appendix A [39]. PCR amplified products (~300 bp) were purified by using enzymes QIA-quick PCR Purification Kit (Qiagen, Germantown, MD, USA), and NGS amplicon sequencing was performed using the MiSeq NGS (Illumina, San Diego, CA, USA) technology. Sequences were analysed using the AV6 primer and compared to those available in the GenBank database (http://www.ncbi.nlm.nih.gov/, accessed on 27 July 2020).

### 2.3. Antimicrobial Susceptibility Test

The isolates were tested for susceptibility to 8 antibiotics using the Kirby–Bauer method on Mueller–Hinton Agar (MH, Liofilchem, Teramo, Italy), and the clinical breakpoints were determined as recommended by the Clinical Laboratory Standards Institute (CLSI) [40]. The following antimicrobial disks were tested (Liofilchem, Teramo, Italy), (class, disk abbreviation code and concentration in brackets) for all the isolates: Ampicillin (AMP; 10 μg), Amoxicillin + clavulanic acid (AMC; 30 μg), Ceftazidime (CTZ; 30 μg), Cefuroxime (CXM, 30 μg), Gentamicin (GN; 30 μg), Doxycycline (DX; 30 μg), Ciprofloxacin (CIP; 5 μg), Enrofloxacin (ENR; 15 μg). The antibiotics were selected on the basis of standardized therapeutic protocols available for turtles. 

After disks application, plates were incubated at 37 °C for 24 h. The zone diameters were measured and strains were classified as susceptible (S), or resistant (R). *Escherichia coli* strain ATCC 25,922 was used as a quality control.

### 2.4. Statistical Analysis

The data were analysed with Excel 2010. Clinical isolates were compared to healthy isolates from O/C samples for: (*i*) proportion of isolates identified at genus level; (*ii*) proportion of resistance against antibiotics. Statistical analyses were performed with the Chi-square test or Fisher test with a significance level *p* < 0.05, and by using SAS software version 8.2.

## 3. Results

A total of 410 isolates were cultured from the 200 turtles; about 98.5% of the isolates (n = 404) were Gram negative, while 1.5% of the isolates (n = 6) were Gram positive. All the Gram negative isolates grew on CBA and MCK media after 24 h of incubation. Gram positive isolates grew on CBA and/or MSA. Three hundred and ninety-five bacteria obtained in purity on plates were bio-enzymatically tested and identified with identity scores higher than 98%. Fifteen isolates were identified by means of 16S rRNA PCR amplification and sequencing. 

### 3.1. Subsections

#### 3.1.1. Identification of Bacterial Isolates from Oral and Cloacal Samples

From the 98 healthy turtles subjected to O/C sampling, 280 Gram negative isolates were cultured and were found to belong to 11 different genera: 65/280 *Aeromonas* (23%), 42/280 *Pseudomonas* (15%), *Vibrio* (15%), 42/280 *Serratia* (15%), 23/280 *Proteus* (8%), 11/280 *Alcaligenes* (4%), 11/280 *Moraxella* (4%), 11/280 *Citrobacter* (4%), 11/280 *Kluyvera* (4%), 11/280 *E. coli* (4%) and 11/280 *Salmonella* (4%). Figure 2 depicts the genera identified in O/C samples. 

#### 3.1.2. Identification of Bacterial Isolates from Wound Samples

From the injured turtles, a total of 102 samples were collected from: skin wounds (42/102), BAL (35/102), internal biopsies (11/102), carapace wounds (13/102) and plastron wound (1/102). A total of 130 clinical bacterial isolates were cultured and 124 of them were Gram negative while 6 were Gram positive.

Isolates not successfully identified by biochemical tests and subjected to 16SrRNA sequencing were: 8 *Pseudomonas* (4 *P. putida* and 4 *P. putrefaciens*), 4 *Vibrio* (2 *V. fluvialis*, 1 *V. metschnikovii* and 1 *V. vulnificus*), 2 *Citrobacter freundii* and 1 *Klebsiella oxythoca*.

As shown in Figure 3, which describes the genera identified in clinical samples, isolates were found to belong to 12 different genera: 29/130 *Aeromonas* (22.3%), 24/130 *Pseudomonas* (18.5%), 22/130 *Citrobacter* (17%), 20/130 *Vibrio* (15.3%), 9/130 *Acinetobacter* (6.9%), 6/130 *Enterobacter* (4.6%) 4/130 *Proteus* (3.1%), 4/130 *Klebsiella* (3.1%), 4/130 *Bacillus* (3.1%), 4/130 *Morganella* (3.1%), 2/130 *Alcaligenes* (1.5%) and 2/130 *Staphylococcus* (1.5%). 

When comparing the prevalence of genera of both groups, significant differences were found only for *Citrobacter* spp., which were more prevalent in clinical samples (*p* < 0.001). 

Several genera, such as *Serratia*, *Moraxella*, *Kluyvera*, *Salmonella*, or species, such as *E. coli*, were isolated only in healthy animals, while *Acinetobacter* spp., *Enterobacter* spp., *Klebsiella* spp., *Bacillus* spp., *Staphylococcus* spp. and *Morganella* spp., were isolated from the wounds of injured animals.

#### 3.1.3. Summary of AMR Patterns

As shown in Table 2, resistance to at least one antibiotic class was found in all the isolates collected (n = 410). 

Of note, 233 out of 410 isolates (56.8%) were found to be resistant to Ampicillin, 242 out of 410 (59%) resistant to Amoxicillin + clavulanic acid, 300 out of 410 (73.1%) resistant to Ceftazidime, 223 out of 410 (54.3%) resistant to Cefuroxime, 53 out of 410 (12.9%) resistant to Ciprofloxacin, 59 out of 410 (14.3%) resistant to Enrofloxacin, 72 out of 410 (17.5%) resistant to Gentamicin, 161 out of 410 (39.2%) resistant to Doxycycline. 

In particular, 280 isolates collected from O/C samples were found to be resistant to: Ampicillin (126/280; 45%), Amoxicillin + clavulanic acid (146/280; 52%), Ceftazidime (22/280; 7.8%), Cefuroxime (140/280; 50%), Doxycycline (108/280; 38.5%), Gentamicin (22/280; 7.8%) Enrofloxacin (11/280; 3.9%), Ciprofloxacin (0/280; 0%), while the remaining 130 clinical isolates were found to be resistant to: Ampicillin (107/130; 82.3%), Amoxicillin + clavulanic acid (96/130; 73.8%), Ceftazidime (96/130; 73.8%), Cefuroxime (83/130; 63.8%), Doxycycline (53/130; 40.7%), Gentamicin (50/130; 38.4%), Enrofloxacin (48/130; 36.9%), Ciprofloxacin (53/130; 40.7%). Figure 4 shows the percentages of resistance of O/C vs. clinical isolates. In total, 50 out of 410 isolates displayed a Multidrug Resistant (MDR) phenotype, i.e., they were resistant to at least three different classes of antibiotics [41].

Twenty isolates from O/C and 30 isolates from clinical samples were MDR. They were: 15 *Pseudomonas* spp., 11 *Citrobacter* spp., 2 *Bacillus* spp. and 2 *Enterobacter* spp. in clinical samples and 12 *Pseudomonas* spp., 4 *Proteus* spp., 2 *Vibrio* spp. and 2 *Citrobacter* spp. in O/C samples. 

Highly significant differences in antibiotic resistance were found between clinical vs. O/C isolates; in particular for AMP (*p* value 2.75 × 10^−12^), AMC (*p* value 5.11 × 10^−5^), CTZ (*p* value 3.20 × 10^−42^), CXM (*p* value 0.011), ENR (*p* value 2.68 × 10^−18^), GN (*p* value 1.01 × 10^−13^) (Chi square test *p* < 0.001). The levels of resistance against CIP were analysed by Fisher test and clinical isolates resulted more resistant than isolates from O/C samples (*p* < 0.001). The rates of resistance against DXT were not significantly different between the two groups of isolates (*p* > 0.05).

## 4. Discussion

Several studies based on traditional culture methods have assessed the composition of microbiota and the antimicrobial susceptibility of bacterial isolates from healthy [27,31,32,33] and injured loggerhead sea turtles [30]. Two common features that emerged from the above studies were: (*i*) Gram negatives represent the predominant bacteria in different samples (i.e., cloacal, eggs, faeces, etc); (*ii*) these isolates usually display high percentages of resistance against antibiotics [26,30,42]. The present report confirmed the isolation of several Gram negative genera in both healthy and injured loggerhead sea turtles, with the majority of isolates belonging to *Enterobacteriaceae* or to fermentative Gram negative organisms such as *Aeromonas* spp., *Pseudomonas* spp., *Acinetobacter* spp. and *Vibrio* spp., [24,43,44,45,46]. Many of these genera, including *Salmonella* spp., have been isolated from water samples collected from sea turtles’ environments [46]. 

No significant differences were found when the percentages of bacterial genera were compared in healthy vs. injured turtles, except for the genus *Citrobacter,* which was more prevalent in clinical samples. 

It is interesting to note that in this study Gram positive bacteria were only found in samples collected from wounds, while they were previously reported also in healthy animals [31,44,46,47,48,49]. 

Bacterial infections seem to be one of the major causes of infectious diseases in sea turtles with high rates of morbidity and mortality [50,51,52] and with Gram negative aerobic bacteria being responsible for the majority of infections [24]. *Vibrio algynolyticus*, *Aeromonas hydrophila*, *Flavobacterium* spp., *Pseudomonas* spp. and *Acinetobacter* spp. are the most common microorganisms isolated from the lesions of sea turtles [24], with serious health consequences including pneumonia, stomatitis, rhinitis, tracheitis, osteomyelitis [53], ulcerative dermatitis, renal diseases, systemic diseases [54] and sepsis [24,55]. In our study, the same genera were isolated from healthy and injured animals, confirming that those organisms may act as opportunistic pathogens, depending on concurrent factors such as parasitic, infectious [24] or traumatic injuries [21,56]. Most of the turtles recruited in the study were caught by local fisherman and therefore they were exposed to trawling risks [18] such as pulmonary problems (i.e., gas embolism) [57]. In fact, 35 out of 102 injured turtles were found with compromised pulmonary status, and this frequency was consistent with previous data which reported that one third of the turtles accidentally caught by trawling nets had pulmonary lesions [57]. We also found 56 out of 102 injured turtles with external wounds, which were probably traumatic since it is known that trawling frequently causes traumatic wounds [58].

More microbial species were found in the O/C swabs than in swabs collected from lesions (in which one or two microbial species were usually isolated per each turtle). Indeed, it has been reported that the environmental factors, the determinants of virulence, and the antimicrobial pressure can modify the prevalence of bacterial composition of injured tissues [59].

The antimicrobial resistance of bacteria from wild animals that have never been subjected to antibiotic therapy represents another interesting finding. All the turtles were sampled soon after their arrival at the STC and, at least to our knowledge, none of them had been previously housed in other rehabilitation centres, as sea turtles are generally marked with a tag attached to the rear fin before being released into the sea [60].

In the present study, all the isolates collected from both groups of turtles were found to be resistant to at least one antimicrobial category, with very high resistance percentages, in particular against β-lactams antibiotics such as Ampicillin, Amoxicillin + clavulanic acid and Ceftazidime. Furthermore, despite having tested only four antibiotic categories, MDR strains were still found and they were more frequent in clinical samples.

Other authors have reported that bacteria isolated from free-ranging healthy sea turtles usually show antimicrobial susceptibility [4,24], particularly against β-lactams, nevertheless, as for other bacterial populations, the AMR phenomenon is constantly evolving and needs to be monitored. Therefore, the role of sea turtles as a reservoir of AMR strains should be investigated deeply and particular attention should be given to subpopulations of sea turtles [4,26,30,61].

The Mediterranean subpopulation of sea turtles is poorly connected with the Atlantic one and, for their site fidelity, juvenile and adult subjects of *C. caretta* are considered to be optimal ecological indicators of this zone. Therefore, studying both juvenile and adult subjects, as in the present report, may be considered a mirror of a specific marine ecological environment [18,26,30].

The constant monitoring of a subpopulation can intercept rapidly an increase in AMR, and could help to investigate better the anthropogenic antimicrobial resistance acquired by free-ranging sea turtles [30,31,46].

Although Gram negative bacteria may exhibit a natural resistance mechanism against β-lactams, when subjected to antibiotic pressure β-Lactamases and Extended-spectrum β-lactamases (ESBLs) tend to evolve to protect these bacteria from β-lactams molecules [62]. These enzymes are some of the most problematic hurdles from a therapeutic point of view. They are widely distributed among livestock and domestic animals as well as humans [62], and drastically reduce the range of therapeutic options. Moreover, these determinants are often located on plasmids carrying resistance genes to other non-β-lactam-antibiotics [62]. ESBLs are widely distributed in farm animals while there are several reports about their presence in several wild species such as wild birds, wild rodents, red foxes (*Vulpus vulpus*), lynx (*Lynx paradinus*), and wild boar (*Sus scrofa*) [63]. High resistance rates against β-lactams antibiotics have also been reported in marine species such as in sea lions (*Zolaphus californianus*), harbour seals (*Phoca vitulina*), elephant seals (*Mirounga augustirostris*) [8]. Focusing on the Mediterranean zone, several reports have described high resistance rates against β-lactams antibiotics, associated with ESBLs-producing bacteria in urban rivers [63], marine waste water [12] and injured loggerhead sea turtles [30]. A recent paper reported that the ESBLs genes identified in wild animals (i.e., loggerhead sea turtles), may be the same as those described in human isolates [30], therefore calling the attention on interspecies transmission mediated by successful ESBLs-producing clones and plasmids among humans, domestic and wildlife species [62,63,64]. The natural resistance mechanisms and the remarkable ability to acquire resistance genes from the environment exhibited from Gram negatives could explain the higher number of MDR strains isolated from *Pseudomonas* spp., *Citrobacter* spp. and *Proteus* spp. Indeed, *Enterobacteriaceae* are considered responsible for the spreading of resistant genes across environmental, animal and human microorganisms [26]. 

The differences in resistance rates were also very important for the class of fluroquinolones. We tested two fluoroquinolones such as Ciprofloxacin (for human use only) and Enrofloxacin (for veterinarian exclusive use). We tested two molecules of this class because it is well documented that the resistance against fluoroquinolones is closely related with the specific classes of molecules and the single molecule cannot mirror the entire category. Moreover, fluoroquinolones are one of the few therapeutic options for sea turtles [65,66,67,68].

We recorded about 40% of resistance in clinical isolates against Ciprofloxacin and Enrofloxacin, in contrast to the very low percentage found in the O/C isolates. Enrofloxacin is recommended for use in *C. caretta* for various infections such as conjunctivitis and lower respiratory diseases due to its long half-life and high bioavailability with low side effects [65]. Our study suggests that treatment with fluoroquinolones in loggerheads needs prior in vitro susceptibility testing, due to the increasing percentage of resistance found in clinical isolates of this species.

Regarding the aminoglycosides, the high percentage of resistance in clinical isolates is in line with a recent study by Chuen-Im and coauthors [68], but is in contrast with other findings [4,26]. Currently, aminoglycosides are used to treat Gram negative infections (such as *Acinetobacter baumannii*, *Enterobacteriaceae*, *Escherichia coli*, *Klebsiella pneumoniae*, and *Pseudomonas aeruginosa*) in human medicine or as a second choice of treatment for MDR bacteria. The use of these antimicrobials in sea turtles is not frequent and only preliminary information is available [24]; thus, further investigation is needed in order to tailor a targeted therapy in wild marine species.

Sea turtles are often considered to be an extremely resilient species, capable of contrasting a plethora of physical, chemical and biological insults [17]. The level of AMR rates reported from different studies on sea turtle populations is extremely variable among countries [68,69]. Therefore, antimicrobial susceptibility testing is recommended to guide the treatment of any bacterial infections [24]. Several studies have demonstrated the direct transmission of resistance determinants between humans and animals; therefore, the role of human activities in spreading AMR determinants to sea turtles deserves further investigation. Systematic surveillance on this sentinel species should be implemented as well.

## 5. Conclusions

In conclusion, a large number of free-living *Caretta caretta* from the Adriatic Sea were screened for AMR over a long period of time and their potential role in spreading antibiotic resistance determinants in the marine ecosystem was hypothesized. The results from this study may help clinicians (either working in rehabilitation centres or in clinics) to set proper antibiotic therapies. They should also be aware that sea turtles can act as carriers of AMR bacterial strains [47]. The presence of high AMR rates in zoonotic bacteria should encourage further research on the impact of anthropogenic pressures (i.e., climate change, pollutants), on sea turtles living in the Adriatic Sea. In fact, only an integrated One Health approach would allow us to design future studies on the link between sea turtle and human health within their shared environment.

## Figures and Tables

**Figure 1 animals-11-02435-f001:**
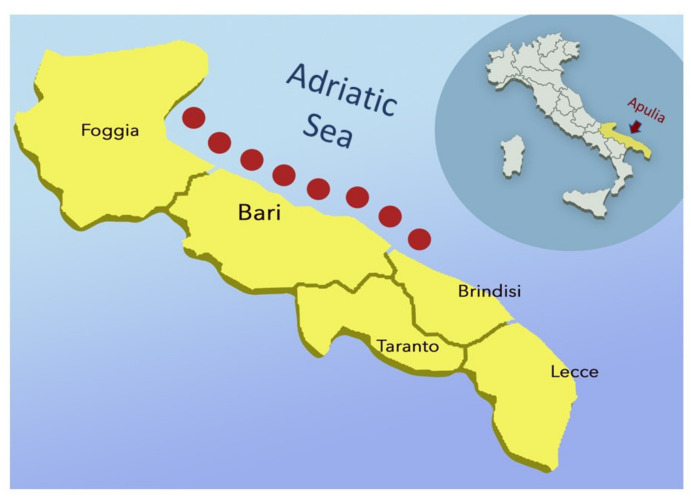
Map of sites (red dots) of the western Adriatic Sea near Apulia (yellow) where the loggerhead sea turtles were found stranded or captured by local fishermen. Image source: www.freepik.com (Accessed date 15 December 2020).

**Figure 2 animals-11-02435-f002:**
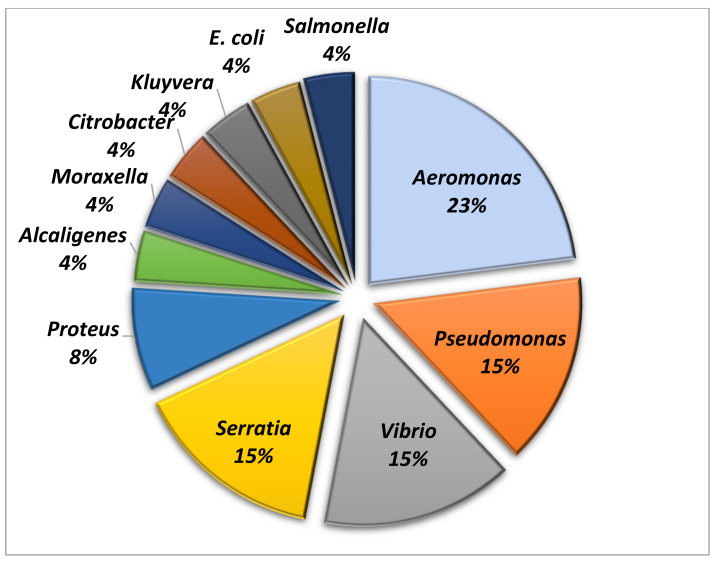
Genera found in oral/cloacal samples.

**Figure 3 animals-11-02435-f003:**
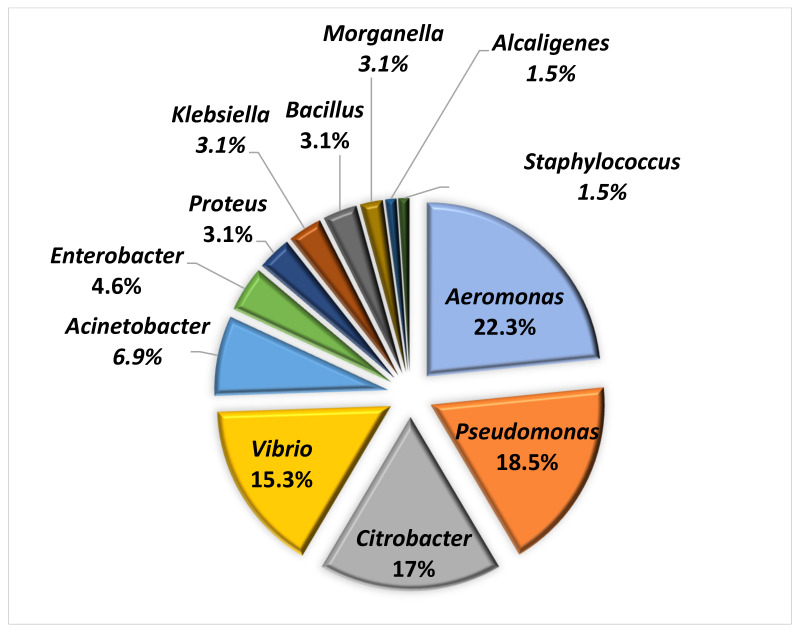
Distribution of genera found in clinical samples.

**Figure 4 animals-11-02435-f004:**
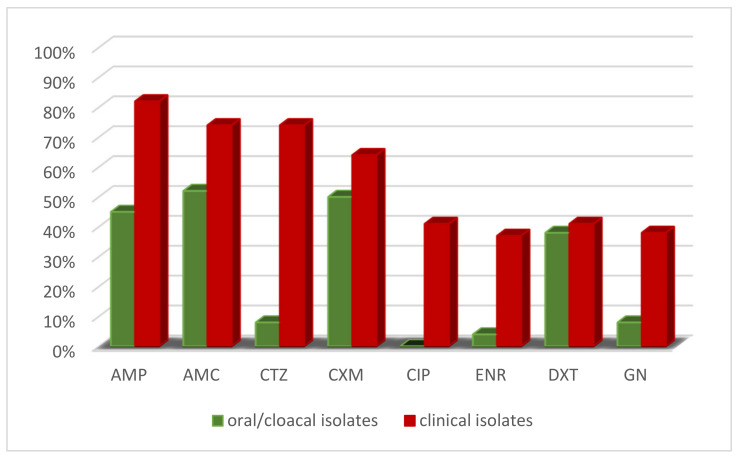
Percentage of AMR of O/C isolates vs. clinical isolates. AMP: Ampicillin, AMC: Amoxicillin + Clavulanic acid; CTZ: Ceftazidime; CXM: Cefuroxime; CIP: Ciprofloxacin; ENR: Enrofloxacin; DXT: Doxycycline, GN: Gentamycin.

**Table 1 animals-11-02435-t001:** Major features of healthy and injured loggerhead sea turtles included in the study (200 subjects).

	Juvenile	Subadult	Adult
Number of subjects	100	76	24
CCL ^a^	31 ± 9.2	53 ± 2.3	74 ± 10.3
Sex ^b^	/	/	11 males/13 females
Weight ^c^	22 ± 8.7	37 ± 6.3	45 ± 2.4
Healthy animals	70	23	5
Injured animals	30	53	19

^a^ CCL, curved carapace length expressed in cm; mean ± standard deviation. ^b^ If applicable because young subjects did not show sexual dimorphism. ^c^ Value expressed in kg; mean ± standard deviation.

**Table 2 animals-11-02435-t002:** Summary of the resistance patterns in all bacterial isolates (n = 410) and percentages of resistance in O/C or clinical isolates.

Antibiotics	Phenotypic Profiles in 410 Isolates		Percentage of Resistance in Oral/Cloacal and Clinical Isolates	
	R	R (%)	O/C Isolates (280) R (%)	Clinical Isolates (130) R (%)
AMP	233 *	56.8 **	45	82
AMC	242	59	52	73.8
CTZ	300	73.1	7.8	73.8
CXM	223	54.3	50	63.8
CIP	53	12.9	0	40.7
ENR	59	14.3	3.9	36.9
DXT	161	39.2	38.5	40.7
GN	72	17.5	7.8	38.4

R, resistant; R (%), resistance percentage. AMP: Ampicillin, AMC: Amoxicillin + Clavulanic acid; CTZ: Ceftazidime; CXM: Cefuroxime; CIP: Ciprofloxacin; ENR: Enrofloxacin; DXT: Doxycycline, GN: Gentamycin. * Number of isolates showing that phenotype. ** Percentage of isolates showing that phenotype.

## Data Availability

Data is contained within the article or Appendix A.

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
