# Peer review of "Antimicrobial Resistance in Loggerhead Sea Turtles (Caretta caretta): A Comparison between Clinical and Commensal Bacterial Isolates"

_animals, 2021, doi:10.3390/ani11082435_

Round 1

Reviewer 1 Report

The paper presents an interesting study on a large number of loggerhead sea turtle specimens sampled over a period of 4 years, which were subjected to swabbing of tissues and analysis of Gram-negatives for antimicrobial resistance. The paper is well conceived and hypothesis is well documented.

Some issues that need to be clarified:

  • Sea turtles are migratory species. Some of the specimens you collected must have been tagged. Did you detect their migratory pattern? If migrating outside of the Western Adriatic Sea, which is very likely, you might need to adjust the last sentence in the Introduction.
  • chapter 2.2. - did you supplement agars with NaCl to adjust salinity for the marine isolates? 
  • Did you try to incubate the inoculated media on lower temperatures more suitable for marine isolates?
  • Have you adapted the API tests for marine isolates i.e. increased the incubation time to 72 hours, lowered the incubation temperature and used a suspension of 1.5% saline as inoculum,  etc? These actions allow for a better overall identification results when testing marine bacteria.
  • which isolates were not successfully identified by API tests and needed the 16SrRNA sequencing? Please indicate.
  • Figures 2,3 - A large proportion of identified bacteria are aeromonads, also Vibrio. However, motile aeromonads are often interpreted as V. vulnificus or A. hydrophila when using API tests. Did you take that into consideration?
  • I suppose you needed to test pseudomonads with 16SrRNA since on API tests they react weakly on sugar fermentation. Please indicate.
  • How likely it is that the injured turtles were previously treated with antibiotic therapy, although without tags? Are there any databases linking such info that might have been entered by veterinary clinics?
  • Minor comments: for affiliations, as far as I can see, there are only two (1 an 2). No need to repeat the name of the same institution (3-7)
  • please re-read the paper to correct inconsistencies and errors, some spelling mistakes, minuscules and italics throughout

Author Response

Comments and Suggestions for Authors

The paper presents an interesting study on a large number of loggerhead sea turtle specimens sampled over a period of 4 years, which were subjected to swabbing of tissues and analysis of Gram-negatives for antimicrobial resistance. The paper is well conceived and hypothesis is well documented.

A: Thank you for your comment and suggestions.

------------------------------------------------------------------------------------------------------------------------

Some issues that need to be clarified:

Q1: Sea turtles are migratory species. Some of the specimens you collected must have been tagged. Did you detect their migratory pattern? If migrating outside of the Western Adriatic Sea, which is very likely, you might need to adjust the last sentence in the Introduction.

Q2: chapter 2.2. - did you supplement agars with NaCl to adjust salinity for the marine isolates? 

Q3: Did you try to incubate the inoculated media on lower temperatures more suitable for marine isolates?

Q4: Have you adapted the API tests for marine isolates i.e. increased the incubation time to 72 hours, lowered the incubation temperature and used a suspension of 1.5% saline as inoculum, etc? These actions allow for a better overall identification result when testing marine bacteria.

Q5: which isolates were not successfully identified by API tests and needed the 16SrRNA sequencing? Please indicate.

A5: Accordingly, the isolates not successfully identified by API test which needed the 16S sequencing have been specified in the manuscript (Please, see lines 242-245)

Q6: Figures 2,3 - A large proportion of identified bacteria are aeromonads, also Vibrio. However, motile aeromonads are often interpreted as V. vulnificus or A. hydrophila when using API tests. Did you take that into consideration?

Q7: I suppose you needed to test pseudomonads with 16SrRNA since on API tests they react weakly on sugar fermentation. Please indicate.

A7: Accordingly, we have specified this in the text (please see lines 242-245), moreover not all pseudomonads needed to be tested with 16S because we also used presumptive identification criteria such as the characteristic fruits-like smell and green pigmentation.

Q8: How likely it is that the injured turtles were previously treated with antibiotic therapy, although without tags? Are there any databases linking such info that might have been entered by veterinary clinics?

A8: Unfortunately, there are no such databases for sea turtles which can be entered by veterinary clinics, but if turtles are untagged, it means that they have never been caught before and therefore never treated with drugs. For this specific reason, we have not included in the study tagged turtles as stated on lines 116-117.

Minor comments:

Q9: for affiliations, as far as I can see, there are only two (1 an 2). No need to repeat the name of the same institution (3-7).

A9: Accordingly, we have deleted the repeated affiliations.

Q10: please re-read the paper to correct inconsistencies and errors, some spelling mistakes, minuscules and italics throughout

A10: Thank you for the suggestion.

The paper has been read carefully and the errors corrected.

Reviewer 2 Report

The objectives of the study are vague. Please specify as necessary. Also, the initial hypothesis of the authors should be presented.

Table 1. Please do not use vertical lines.

Table 1. What type of injuries were present in these animals?

Swab culture. Why only up to 48 h and to up to 72 h, which is a better protocol?

The PCR conditions, primers etc. should be described in a supplementary table.

Why did you use the disk method and not a more advanced one, e.g., Vitek? Please justify.

Did you define multi-resistant isolates? How?

The bacterial genera isolated should be presented in Tables, not in the text. Why not identification at species level? Please justify.

Table 2. These are not AMR patterns, this a summary of the patterns. The patterns must be described in detail in supplementary material.

Table 3 is too detailed and can be safely deleted altogether.

There is a need to present details about multi-resistant isolates. This is a serious omission.

The discussion is ok, but lacks some recent relevant references.

The conclusions is very generalistic. Either correct or omit altogether.

Overall.
Extensive revision and re-evaluation. 

Author Response

Reviewer 2:

Comments and Suggestions for Authors

Q1: The objectives of the study are vague. Please specify as necessary. Also, the initial hypothesis of the authors should be presented.

A1: Thank you for the suggestion. Accordingly, the text has been modified (Please see lines 96-100)

Q2: Table 1. Please do not use vertical lines.

A2: Accordingly, the table has been modified. 

Q3: Table 1. What type of injuries were present in these animals?

A3: Accordingly, we have specified in the text what types of injuries the animals presented (Please see lines 152-157). We have not included these data in Table 1 to avoid redundancies. In addition, we have provided further details on lesions sampled (please see lines 158-162).

Q4: Swab culture. Why only up to 48 h and to up to 72 h, which is a better protocol?

 A4: We used validated protocols which allow to identify well bacaterial species from sea turtles as described by other authors in this field (Foti et al., 2009; Pace et al., 2019; Alduina et al., 2020; Blasi et al., 2020). 

Q5: The PCR conditions, primers etc. should be described in a supplementary table.

A5: Thank you for the suggestion. Accordingly, PCR conditions and primers have been described in a supplementary table.

Q6: Why did you use the disk method and not a more advanced one, e.g., Vitek? Please justify.

A6: We use validated protocols that allow to optimally detect the resistance of bacterial isolates, as described also by other authors (Alduina et al., 2020; Blasi et al., 2020; Chuen-Im et al., 2021).

Q7: Did you define multi-resistant isolates? How?

A7: Accordingly, the definition of MDR phenotype i.e., resistance to at least three different classes of antibiotics (Magiorakos et al., 2012), has been added in the manuscript (please see lines 290-295).

Q8: The bacterial genera isolated should be presented in Tables, not in the text. Why not identification at species level? Please justify.

A8: Thank you for the comment.

We included data on bacterial identification both in the text and in two pie charts (please see Figure 2 and 3), since they both allowed a better visualization and interpretation of the data. In particular, the pie chart is, in our opinion, the optimal choice when showing how individual data make up the whole set of data.

For bacterial identification we preferred the biochemical methods given the large number of bacteria that were isolated and identified. However, with the biochemical identification it is possible to identify precisely the genera but not the species, in particular for isolates from sea turtles. Therefore, since this is an epidemiological study designed to compare the resistance patterns, we did not sequence all the isolates in order to achieve the species identification.

Q9: Table 2. These are not AMR patterns, this a summary of the patterns. The patterns must be described in detail in supplementary material.

A9: We agree with the Reviewer, these are summary of patterns and we have therefore corrected the Table 2 caption (Please see line 268). This was a large and quite long study whose aim was to describe and summarize the AMR patterns observed in these two groups of samples. Thus, it was not the scope of this manuscript to screen all the strains with an increased AMR rate; in fact, this is the focus of our current and further investigations.

Q10: Table 3 is too detailed and can be safely deleted altogether.

A10: Thank you for the suggestion.

Accordingly, Table 3 has been deleted and the respective p-values have been added in the text (please see lines 306-308)

Q11: There is a need to present details about multi-resistant isolates. This is a serious omission.

A11: Thank you for the suggestion.

Accordingly, we have added details about MDR strains (Please see lines 292-295 and the Discussion section lines 365-367)

Q12: The discussion is ok, but lacks some recent relevant references.

A12: Accordingly, we have updated the references (Please see reference number 35,36,41,68,69).

Q13: The conclusions is very generalistic. Either correct or omit altogether.

A13: Thank you for the comment.

Accordingly, the conclusions have been deleted and the last paragraph of the Discussion, has been revised (Please see lines 446-456).

Reviewer 3 Report

Introduction

First paragraph is redundant. These are well known facts.

Objectives are not defined clearly

Materials and methods

Table 1 is formatted badly.

Please define juvenile, sub-adult, adult by describing the exact age separation in months.

All the details of PCR should be transferred in a supplementary material, not in the text.

Please describe the standards employed for definition of susceptibility / resistance.

Statistics. Not percentage, proportion is the correct word.

Results

A statistical comparison of bacterial frequencies recovered from injury samples and healthy animal samples will add value to the manuscript.

Discussion

The discussion is verbose and lengthy and should be reduced.

Also, there is no mention about multi-resistant strains.

Finally, please add two sentences about the one-health implications of the findings.

General

The work can be published but needs extensive corrections.

Author Response

Reviewer 3:

Comments and Suggestions for Authors

-Introduction

Q1: First paragraph is redundant. These are well known facts.

A1: Thank you for the suggestion.

Accordingly, the introduction has been modified, please see lines 48-52.

Q2: Objectives are not defined clearly

A2: Accordingly, the objectives have been clarified, please see lines 96-100.

-Materials and methods

Q3: Table 1 is formatted badly.

A3: Accordingly, the table has been modified. 

Q4: Please define juvenile, sub-adult, adult by describing the exact age separation in months.

A4: Thank you for the question.

We decided to classify the animals by parameters such as curved carapace length (CCL), and weight (lines 117-121) since Musick and Limpus, in 1997, proposed the following classification for marine turtles:

i)               Hatchlingor post-hatchling: baby turtles (0.02-1 kg) in their first year from the terrestrial to first period into pelagic phase;

ii)             Juveniles: young turtles (1-25 kg) from 1 to 10 years old and not yet sexually mature;

iii)            Sub-adult: turtles (25-90 kg), from 10 to 30 years old living into neritic environments and close to the beginning of their reproductive period;

iv)            Adults: turtles (60-120 kg), from 45 up to 60+ years old with clear sexual dimorphism.

The life stage classification, which is rather empiric, cannot be precise as a large variability in growth rates are observed in sea turtles due to genetical, sexual and environmental factors (Bjondal et al., 2002; Hepper et al., 2003; Casale et al., 2011). For comparative studies in turtles in rehabilitation centers, biometrical and morphological studies are commonly used (Casale et al., 2018). CCL). Therefore, curved carapace width (CCW), straight carapace length (SCL), and turtle weight are usually compared to define the precise life-stage of the animal. When using this method, the adults are considered ready for reproduction when they reach the size of at least 60-75 cm CCL and up to 40 kg weight (Casale et al., 2005; 2006; Hayes et al., 2010); at this life stage they show the main sexual dimorphism character: the tail length (Bolten, 1999). 

Q5: All the details of PCR should be transferred in a supplementary material, not in the text.

A5: Thank you for the suggestion. Accordingly, PCR conditions and primers have been described in a supplementary table.

Q6: Please describe the standards employed for definition of susceptibility / resistance.

A6: The halos generated by the isolates on agar medium were measured and compared with the CLSI reference standards (Please see lines 196-199).

Q7: Statistics. Not percentage, proportion is the correct word.

A7: Accordingly, the words have been changed (Please see lines 212-213).

-Results

Q8: A statistical comparison of bacterial frequencies recovered from injury samples and healthy animal samples will add value to the manuscript.

A8: A statistical comparison of bacterial frequencies observed in healthy and unhealthy samples are reported in the manuscript, please see lines 256-258.

-Discussion

Q9: The discussion is verbose and lengthy and should be reduced.

A9: Thank you for the suggestion, accordingly, the discussion has been modified.

Q10: Also, there is no mention about multi-resistant strains.

A10: Accordingly details about MDR isolates have been added in the manuscript (Please see lines 292-295, and the Discussion (Please see lines 365-367).

Q11: Finally, please add two sentences about the one-health implications of the findings.

A11: Accordingly, sentences about the One-health implications have been added (Please see lines 451-455).

Round 2

Reviewer 2 Report

The authors have responded well to the comments and have made changes, which improve the manuscript.

Before, final acceptance, please include a new supplementary table listing in tabular form the type of injuries present in the animals of the study (comment 3 of the previous review).
So, minor revision and after making the requested change, the manuscript can be accepted.

Author Response

Answers to Reviewers’ comments.

Reviewers n°2

The authors have responded well to the comments and have made changes, which improve the manuscript.

Before, final acceptance, please include a new supplementary table listing in tabular form the type of injuries present in the animals of the study (comment 3 of the previous review).
So, minor revision and after making the requested change, the manuscript can be accepted.

A1: Thank you for the suggestion.

Accordingly, the Table 1. summarizing the type of injuries has been added in the Supplementary materials.

Reviewer 3 Report

The manuscript is improved and is almost ready for publication.
Two points that need attention are the following.
1. The definition of age of the animals must be included in supplementary material to help future readers. It is described very well, just add in a supplementary text.
2. Why did the authors use CLSI standards and not the EUCAST standards, since they are in Europe? Also, there are now standards specifically for veterinary work issued jointly by EFSA and EUCAST. Employing those, will add value to their work.

In general, this is a useful study and after making these small changes, the paper can be accepted.

Author Response

 Reviewers n°3

The manuscript is improved and is almost ready for publication.
Two points that need attention are the following.

Q1: The definition of age of the animals must be included in supplementary material to help future readers. It is described very well, just add in a supplementary text.

A1: Thank you for the comment.

For the classification of the age of turtles, we have referred to previously published criteria, which were acknowledged in the paper (reference n.18, 35, 36 and many other are also listed below. We think that including such criteria in the Supplementary Materials would not be proper as they were described and published by other Authors and not by us. Therefore, to avoid plagiarism, the table proposed by the Referee has not been included in the manuscript, but the reader could find any information in the cited papers.

-Bolten A.B. (1999). Techniques for measuring sea turtles. In: Research and Management Techniques for the Conservation of Sea Turtles. (Eds.) Eckert K.L., Bjorndal K.A., Abreu-Grobois F.A., Donnelly M. IUCN/SSN Marine Turtles Specialist Group, Washington D.C. pp. 110-114. 

-Bjorndal K.A, Bolten A., Martins H.R. (2002). Somatic growth model of juvenile loggerhead sea turtles Caretta caretta: duration of pelagic stage. Mar. Ecol. Prog. Ser. 202;265-272.

-Heppell S.S., Snower M.L., Crowder L.B. (2003): Sea turtle population ecology. In: The Biology of Sea Turtles (Eds.) Lutz P.L., Musick J.A., Wyeneke J. CRC Marine Biology Series, CRC Press, Boca Raton. Vol II. pp. 275-306.

-Casale P., Mazaris A., Freggi D. (2011). Estimation of age at maturity of loggerhead sea turtles Caretta caretta in the Mediterranean using length-frequency data. Endang. Species Res. 13;123-129. http://doi.org/10.3354/esr00319.

-Casale P., Broderick A., Camiñas J., Cardona L., Carreras C., Demetropoulos A., Fuller W., Godley B., Hochscheid S., Kaska Y., et al. (2018). Mediterrananean sea turtles: current knowledge and priorities for conservation and research. Endanger. Species Res. 36;229-267. http://doi.org/10.3354/esr00901.

-Casale P., Lazar B., Pont S., Tomás J., Zizzo N., Alegre F., Badillo J., Di Summa A., Freggi D., Lackovic G., et al. (2006). Sex ratios of juvenile loggerhead sea turtles Caretta caretta in the Mediterranean Sea. Mar. Ecol. Prog. Ser. 324;281-285. http://doi.org/10.3354/meps324281.

-Casale P., Freggi D., Basso R., Argano R. (2005). Size at male maturity, sexing methods and adult sex ratio in loggerhead turtles (Caretta caretta) from Italian waters investigated through tail measurements. Herpetol. J. 15(3);145-148.

-Hays G.C, Fossette S., Katselidis K.A., Schofield G., Gravenor M.B. (2010). Breeding periodicity for male sea turtles, operational sex ratios, and implications in the face of climate change. Biol. Conserv. 24;1636-1643. http://doi.org/0.1111/j.1523-1739.2010.01531.x.

Q2: Why did the authors use CLSI standards and not the EUCAST standards, since they are in Europe? Also, there are now standards specifically for veterinary work issued jointly by EFSA and EUCAST. Employing those, will add value to their work.

A2: Thank you for the question and suggestion.

We are aware that there are differences between CLSI and EUCAST disk-diffusion AST guidelines.

CLSI clinical breakpoints, which are based on pharmacokinetic and pharmacodynamic information of the antibiotics in domesticated animal species, are recommended for clinical isolates i.e., when a targeted treatment is needed.

For this reason, and in the absence of established clinical breakpoints for wild marine isolates, we decided to use the clinical cut-offs recommended by CLSI.  In our study, a large number of animals were sampled because they were sick and needed antibiotic treatment to heal internal or external lesions; the CLSI standards were optimal to evaluate the strain response to the antibiotics and to help clinicians set the best treatment (Ahasan et al., 2017a; Turnidge and Paterson, 2007).

In addition, numerous authors such as Alduina et al., 2020, Blasi et al., 2020, Foti et al., 2009, Chen-Im et al., 2021, etc. have also used the CLSI breakpoints.
